# TOWARD HUMAN-INTERPRETABLE EXPLANATIONS IN A UNIFIED FRAMEWORK FOR GNNS

## ABSTRACT

As Graph Neural Networks (GNNs) are increasingly applied across various domains, explainability has become a critical factor for real-world applications. Existing post-hoc explainability methods primarily focus on estimating the importance of edges, nodes, or subgraphs in the input graph to identify substructures crucial for predictions. However, these methods often lack human interpretability and do not provide a unified framework that incorporates both model-level and instance-level explanations. In this context, we propose leveraging a set of graphlets—small, connected, non-isomorphic induced subgraphs widely used in various scientific fields—and their associated orbits as human-interpretable units to decompose GNN predictions. Domain experts can select the most relevant graphlets as interpretable units and request unified explanations based on these units. To address this problem, we introduce UO-Explainer, the **U**nified and **O**rbit-based Explainer for GNNs, which utilizes predefined orbits that are generalizable and universal across graph domains as interpretable units. Our model decomposes GNN weights into orbit units to extract class-specific graph patterns (model-level) and to identify important subgraphs within individual data instances for prediction (instance-level). Extensive experimental results demonstrate that UO-Explainer outperforms existing baselines in providing meaningful and interpretable explanations across both synthetic and real-world datasets. Our code and datasets are available at `https://anonymous.4open.science/r/uoexplainer-F12C`.

## 1 INTRODUCTION

Graph Neural Networks (GNNs) have achieved state-of-the-art performance in various domains including real-world graph-structured data, such as social networks (Fan et al., 2019), molecules (Duvenaud et al., 2015), and knowledge graphs (Hogan et al., 2021). Despite the remarkable advancements in GNN architectures (Hamilton et al., 2017; Kipf & Welling, 2017; Xu et al., 2019; Veličković et al., 2018), they are still perceived as black box models due to their lack of explainability. This deficiency limits trust in GNN predictions, hindering their application in areas such as drug development (Gaudelet et al., 2021) and education (Nakagawa et al., 2019). Therefore, interpreting the prediction of GNNs has become crucial and has led to the emergence of various explanatory approaches.

Explainability methods in graph domains deliver explanations through subgraphs that play a significant role in predictions regarding the input graph. Two primary issues arise regarding explainability: i) a human-interpretability and ii) a unified framework encompassing both model and instance levels. Many existing post-hoc explanability is grounded on perturbation-based methods (Ying et al., 2019; Luo et al., 2020; Xie et al., 2022; Zhang et al., 2023; Schlichtkrull et al., 2022) and gradient-based methods (Baldassarre & Azizpour, 2019; Pope et al., 2019) to approximate the importance of edges or nodes within subgraphs. This stochastic optimization of importance is computationally effective in applying any graph-structured data, these methods have the potential risk that the output subgraph does not align with prior human assumption or knowledge. Specifically, consider a scientist studying gene networks who is interested in understanding whether the presence of certain structures, such as triangles or rectangles, plays a crucial role in predicting specific properties. This scientist would want to audit the model's predictions by utilizing an explanation method that can highlight the importance of these structures. With existing methods, it is challenging to obtain explicit insights into whether a triangular or rectangular structure in the network is more important; instead, users often have to rely

on conjecture based on the generated explanations. Additionally, these explanations do not always present results in a human-intuitive form, as they may include isolated nodes or disconnected edges, further hindering their interpretability. Therefore, we emphasize the need for explainability that centers on human interpretability, a perspective that has been largely underexplored.

Last but not least, human-interpretable explanations should be delivered in a unified framework, incorporating both global and local levels in terms of the scope of explanation (Yuan et al., 2022; Prado-Romero et al., 2024). Most of the existing methods primarily specialize one one-level explanation either model or instance-level explanations. Model-level methods reveal patterns that GNNs deem significant for specific classes, eventually offering a broad understanding of the GNNs' general behavior. On the other hand, instance-level methods focus on individual predictions, identifying the subgraphs most relevant to the target node or graph. As each type of explanation complements the other from different perspectives, understanding at both levels enhances the explainability necessary for grasping the decision-making process of GNNs. However, State-of-the-art (Azzolin et al., 2022; Chen et al., 2023) studies also have rarely explored a unified framework that simultaneously provides both model-level and instance-level explanations in the user-centric perspective of interest.

In this paper, we prioritize the two crucial perspectives that explanation should be human-interpretable in the unified framework incorporating both model and instance levels. Our proposed model, UO-Explainer, the **U**nified and **O**rbit-based Explainer for GNNs, allows users to harness their prior knowledge by giving room to select the user-defined explanation units in unified views. Considering the uniqueness of the graph domain to define explanation unit, we exploit orbits within graphlets that have been studied as a generalizable and universal pattern in many scientific fields such as protein interaction networks Pržulj et al. (2004); Pržulj (2007), social networks Chen & Lui (2018); Ahmed et al. (2015), and molecular structure networks Kondor et al. (2009), while users can also adopt a unique prior perspective to define their unit instead of orbit. To provide unified explanations, we decompose the weights into orbit representation vectors to understand the contribution of each orbit for a specific class or prediction. When we break down the weight into orbits, we acknowledge the contribution of each orbit for model decision-making, which is supposed to be further discussed in the method section in detail. Through extensive experiments, we demonstrate the superior performance of UO-Explainer on eight well-known datasets. Consequently, UO-Explainer shows the concrete performance in a unified framework leveraging the human prior knowledge as orbit generalizable unit in graph-structured datasets.

In summary, the contributions of our research are as follows:

- We propose UO-Explainer, a unified framework to provide both model-level and instance-level explanations for node classification based on human-defined explanation units.

- We demonstrate the human interpretable explanation based on orbits generalizable and effective on graph-structured datasets.

- We perform rigorous and extensive experiments on 8 datasets to evaluate the quality of our explanations in both model and instance-level explanations.

## 2 PRELIMINARY

### 2.1 GRAPH NEURAL NETWORKS (GNNS)

We represents a graph as $\mathcal{G} = (\mathcal{V}, \mathcal{E}; \mathbf{A}, \mathbf{X})$ where $\mathcal{E}$ denotes a edge set and $\mathcal{V} = \{v_1, v_2, \cdots, v_{|\mathcal{V}|}\}$ denotes a node set. $\mathbf{A} \in \mathbb{R}^{|\mathcal{V}| \times |\mathcal{V}|}$ denotes the adjacent matrix and $\mathbf{X} \in \mathbb{R}^{|\mathcal{V}| \times d_{in}}$ denotes the node feature matrix. In this study, we focus on GNNs for node classification tasks as presented in (Kipf & Welling, 2017; Xu et al., 2019). A GNN model $f(\cdot)$ maps input graph into prediction matrix $f(\mathbf{A}, \mathbf{X}) = \mathbf{Z} \in \mathbb{R}^{|\mathcal{V}| \times |\mathcal{C}|}, \mathcal{C} = \{c_1, \cdots, c_{|\mathcal{C}|}\}$ in where $\mathcal{C}$ as the set of classes. GNNs can be expressed as a composite function $f = f_D \circ f_E$ of an embedding-model $f_E(\cdot)$ and a downstream-model $f_D(\cdot)$. The embedding model embeds an adjacent matrix and node feature matrix into a node representation matrix $\mathbf{H}$, i.e., $f_E(\mathbf{A}, \mathbf{X}) = \mathbf{H} \in \mathbb{R}^{|\mathcal{V}| \times d}$. The representation vector of each node $v_n$ is denoted by the $\mathbf{h}_{v_n}$, $n$-th row vector of matrix $\mathbf{H}$. The downstream-model maps the node representation matrix into the prediction matrix to classify nodes into each class, i.e., $f_D(\mathbf{H}) = \mathbf{H}\mathbf{W} + \mathbf{b} = \mathbf{Z}$ where $\mathbf{W} \in \mathbb{R}^{d \times |\mathcal{C}|}$ denotes weight matrix and $\mathbf{b} \in \mathbb{R}^{|\mathcal{C}|}$ denotes bias vectors. In this operation, only the $m$-th column

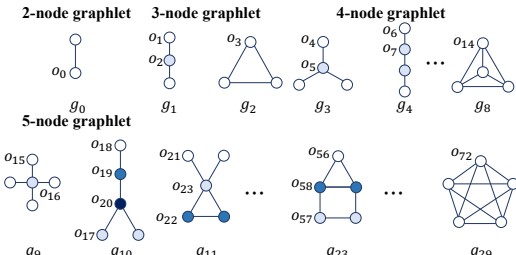

Figure 2: Overview of UO-Explainer: The propagation process of the GNN is depicted in the blue box, while the orange box represents the pipeline of UO-Explainer. The provided explanations are illustrated in the green box. The dark green node of the input graph denotes the target node. The colors (red, orange, blue, and yellow) correspond to the orbits within the graphlets.

vector $\mathbf{w}_{c_m}$ of weight matrix $\mathbf{W}$ and the $m$-th element $b_{c_m}$ of bias vectors $\mathbf{b}$ are involved in the computation to predict the class $c_m$, i.e., $\mathbf{z}_{c_m} = \mathbf{H}\mathbf{w}_{c_m} + b_{c_m}$ where $\mathbf{z}_{c_m}$ denotes the $m$-th row vector of matrix $\mathbf{Z}$. Weight regards to specific class as $\mathbf{w}_{c_m}$ affects only the prediction of the class $c_m$, so we call this weight vector a class weight. Furthermore, the prediction value $z_{v_n,c_m}$ for the class of each node (the $n$-th row and $m$-th column element of $\mathbf{Z}$) can be expressed as $z_{v_n,c_m} = \mathbf{h}_{v_n} \cdot \mathbf{w}_{c_m} + b_{c_m}$, computed by the operation between each node representation vector and class weight.

## 2.2 ORBITS WITHIN GRAPHLETS

Graphlets as predominantly observed patterns are pre-defined subgraphs with a small number of nodes (Pržulj et al., 2004). Figure 1 shows some examples of 2-5 node graphlets (Pržulj et al., 2004; Pržulj, 2007), where $g_l$ denotes the $l$-th graphlet. The full set of 2-5 node graphlets used in our study is presented in Appendix A. All graphlets are non-isomorphic to each other, indicating that each graphlet has a unique structure. Each graphlet contains nodes with identical or distinguishable topological positions known as **orbits**, e.g., the central node of $g_1$ belongs to $o_2$ due to its distinguishable position, whereas the remaining nodes belong to $o_1$. The set of orbits is represented as $\mathcal{O} = \{o_0, \cdots, o_k, \cdots, o_{|\mathcal{O}|}\}$ where $o_k$ denotes the $k$-th orbit.

Figure 1: Graphlets having 2-5 nodes and 0-72 orbits. The same color nodes within each graphlet belong to the same orbit.

Previous studies have highlighted the usefulness of graphlet-based analysis in various graph data domains, including protein interaction networks (Pržulj et al., 2004; Pržulj, 2007), social networks (Chen & Lui, 2018; Ahmed et al., 2015), and molecular structure networks (Kondor et al., 2009). For example, (Pržulj, 2007) defined the Graphlet Degree Distribution (GDD) using 2-5 node graphlets and orbits as units to analyze agreement among biological and chemical networks. (Shervashidze et al., 2009; Espejo et al., 2020) further compared the empirical similarity of various chemical compound networks using a 2-5 node graphlet kernel. Since these analyses indicate that 2- to 5-node graphlets can serve as simple yet effective units for interpreting graph data, we primarily employ them to provide orbit-based explanations, unless otherwise specified by the user.

## 3  UO-EXPLAINER

UO-Explainer serves as a unified explainer capable of delivering both model-level and instance-level explanations for node classification tasks. To deliver explanations in a human-interpretable way, we exploit a pre-defined set of 0-72 orbits as the explanatory unit, which is recognized as essential and human-interpretable units within graph domains, while users can also apply other meaning units regarding their perspective instead of orbits. Upon the interpretable unit, we decompose the weights

that directly influence classification concerning two components such as an embedding model and a downstream task model. An overview of UO-Explainer is presented in Figure 2.

## 3.1 ORBIT BASIS LEARNING

To decompose class weights into orbit units requires orbit bases, which are the representation vectors of each orbit. Orbit bases must necessarily reflect the following two aspects: (1) The distribution of each orbit within the input graph, and (2) The message passing and aggregation behavior of the embedding model. To meet the first requirement, we pre-process the orbit-existences on each node in the input graph. Orbit existence is denoted as $y_{v_n, o_k}$ and determines whether each node $v_n$ belongs to the orbit $o_k$.

$$y_{v_n, o_k} = \begin{cases} 0 & if\ v_n\ doesn't\ belong\ to\ o_k \\ 1 & if\ v_n\ belongs\ to\ o_k. \end{cases} \quad (1)$$

We present the toy example of this pre-processing through Figure 3. (a) portrays the input graph. (b) represents the graphlets and orbits employed in the pre-processing. For simplicity, let us assume that only the graphlets $g_2, g_3, g_6$ are utilized and the orbits used for pre-processing are $o_3, o_4, o_5, o_{11}$. Different color nodes inside each graphlet refer to nodes belonging to different orbits. (c) is a substantial pre-processing process, which checks whether each node can belong to each orbit and assigns 1 or 0 to the orbit's existence. The dark green node represents the pre-processing node. You can see that node 3 belong to $o_3, o_5, o_{11}$. Therefore, $y_{v_3, o_3}, y_{v_3, o_4}, y_{v_3, o_5}, y_{v_3, o_{11}}$ are assigned values of 1, 1, 0, 1, respectively. This pre-processing is performed for all orbits in 2-5 node graphlets at all nodes within the input graph.

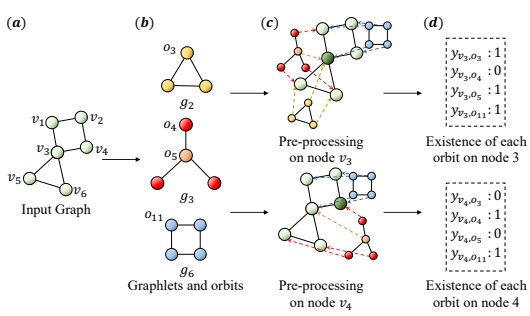

Figure 3: Detailed example of the pre-processing step. The dark green node represents the target node for explanation. The colors of nodes within the graphlets represent the orbits respectively.

Next, we train a logistic binary classifier to predict the existence of each orbit, initializing each orbit with $\hat{\mathbf{p}}_{o_k}$ vector, and taking node representation as input, described by the following equation:

$$\hat{y}_{v_n, o_k} = sigmoid(\hat{\mathbf{p}}_{o_k} \cdot h_{v_n}). \quad (2)$$

To satisfy the second aspect, we learn the orbit basis by incorporating the node representations from the node embedding model when training. Then, we apply normalization in Equation as $\mathbf{p}_{o_k} = \frac{\hat{\mathbf{p}}_{o_k}}{\|\hat{\mathbf{p}}_{o_k}\|}$ to ensure that the size of the orbit basis remains constant. Orbit-basis learning is conducted for all orbits, and details of orbit basis learning can be found in Algorithm 1.

## 3.2 MODEL-LEVEL EXPLANATIONS

Model-level explanations are provided by decomposing class weights into a linear combination of orbit bases, as the following equation:

$$\mathbf{w}_{c_m} \approx s_{c_m, o_0} \mathbf{p}_{o_0} + \cdots + s_{c_m, o_k} \mathbf{p}_{o_k} + \cdots + s_{c_m, o_K} \mathbf{p}_{o_K}$$
$$\approx \sum_{o_k \in \mathcal{O}} s_{c_m, o_k} \mathbf{p}_{o_k}. \quad (3)$$

Generally, when a vector is expressed as a linear combination of bases, the coefficients of each basis indicate to what extent they contribute to forming the vector. Accordingly, the coefficients of orbit bases are regarded as contributions to the class weights. Furthermore, the bases are learned by considering each orbit distribution, thereby treating the contribution of orbit basis as the contribution of each orbit. We define the contribution of orbit $o_k$ to the class $c_m$ classification as a class-orbit score $s_{c_m, o_k}$.

The class-orbit scores are trained by the following objective function derived from Equation 3:

$$\min_{s_{c_m, o_k} > 0} \left\| \mathbf{w}_{c_m} - \sum_{o_k \in \mathcal{O}} s_{c_m, o_k} \mathbf{p}_{o_k} \right\|. \quad (4)$$

---

**Algorithm 1** Orbit Basis Learning

---

**Input:** A set of node representation vectors $\{\mathbf{h}_{v_1}, \cdots, \mathbf{h}_{v_n}, \cdots, \mathbf{h}_{v_{|\mathcal{V}|}}\}$, a set of existences of each orbits for each nodes $\{y_{v_1,o_0}, y_{v_2,o_0}, \cdots, y_{v_1,o_{23}}, \cdots, y_{v_{|\mathcal{V}|},o_{72}}\}$.

**Output:** A set of orbit bases $P = \{\mathbf{p}_{o_0}, \cdots, \mathbf{p}_{o_k}, \cdots, \mathbf{p}_{o_{72}}\}$

**Initialize** $P$ as an empty set

1: **for** k = 0 to 72 (the number of orbits used) **do**
2:      Initialize $\hat{\mathbf{p}}_{o_k}$ as a random vector.
3:      **for** n=1 to $|\mathcal{V}|$ (the number of nodes) **do**
4:          $\hat{y}_{v_n,o_k} = sigmoid(\hat{\mathbf{p}}_{o_k} \cdot \mathbf{h}_{v_n})$
5:          $L = BCE(y_{v_n,o_k}, \hat{y}_{v_n,o_k})$
6:          Update $\hat{\mathbf{p}}_{o_k}$
         via $\nabla_{\hat{\mathbf{p}}_{o_k}} L$
7:      **end for**
8:      $\mathbf{p}_{o_k} = \frac{\hat{\mathbf{p}}_{o_k}}{\|\hat{\mathbf{p}}_{o_k}\|}$
9:      $P.add(\mathbf{p}_{o_k})$
10: **end for**
11: **Return** $P$

---

To consider only the positive impact of the contributing orbits, we limit the class-orbit score to positive. However, directly optimizing the objective function for all orbit bases involves a significant amount of randomness. For example, in the worst case, if all orbit bases are orthogonal, and the dimension of the class weight is $d < |\mathcal{O}|$, an infinite number of combinations of $s_{c_m,o_k}$ can be found to optimize the Equation 4. This randomness hinders the learning of the correct contribution of orbits. Therefore, we modify the objective function using a greedy approach by selecting the orbit that minimizes the difference between the class weight and the linear combination of selected orbits in each iteration, as shown in the following equation:

$$\underset{o_k \in \mathcal{O}}{\arg\min} \ \underset{\mathbf{S}_{c_m} > 0}{\min} \|\mathbf{w}_{c_m} - [\mathbf{P}_{c_m}|\mathbf{p}_{o_k}]\mathbf{S}_{c_m}\|. \tag{5}$$

$\mathbf{P}_{c_m}$ is a matrix consisting of the selected $\mathbf{p}_{o_k}$ as columns, and $\mathbf{S}_{c_m}$ represents a column vector consisting $s_{c_m,o_k}$ of the selected orbits. $[\mathbf{P}_{c_m}|\mathbf{p}_{o_k}]$ denotes concatenation of the $\mathbf{p}_{o_k}$ to $\mathbf{P}_{c_m}$ as a column, e.g., if orbits 1, 3, and 5 are selected, then $\mathbf{P}_{c_m} = [\mathbf{p}_{o_1}|\mathbf{p}_{o_3}|\mathbf{p}_{o_5}]$ and $\mathbf{S}_{c_m}$ is a vector composed of $s_{c_m,o_1}, s_{c_m,o_3}$ and $s_{c_m,o_5}$. By stopping the selection when the difference between the class weight and the linear combination does not decrease, we reduce the randomness and prevent too many orbits from being included in the explanation for each class. The detailed procedure can be found in Algorithm 2.

UO-Explainer uses the orbit $o_{c_m}^*$ with the highest class-orbit score from Equation 6 as the model-level explanation.

$$o_{c_m}^* = \underset{o_k \in \mathcal{O}}{\arg\max} \ \{s_{c_m,o_0}, \cdots, s_{c_m,o_k}, \cdots, s_{c_m,o_{72}}\}. \tag{6}$$

The orbit with the highest class-orbit score is always accompanied by its corresponding graphlet. Therefore, the model-level explanation is provided in the form of graph patterns, with the given orbit as the target node and its corresponding graphlet.

### 3.3 Instance-level Explanations

Instance-level explanations are provided by decomposing the prediction value of the target node into orbit units. The class weight decomposition of Equation 3 extends to the decomposition of prediction values as follows:

$$\begin{aligned} z_{v_n,c_m} &\approx \mathbf{h}_{v_n} \cdot \mathbf{w}_{c_m} + b_{c_m} \\ &\approx \underbrace{s_{c_m,o_0}\mathbf{h}_{v_n} \cdot \mathbf{p}_{o_0}}_{s_{v_n,c_m,o_0}} + \cdots + \underbrace{s_{c_m,o_k}\mathbf{h}_{v_n} \cdot \mathbf{p}_{o_k}}_{s_{v_n,c_m,o_k}} + \cdots + \underbrace{s_{c_m,o_K}\mathbf{h}_{v_n} \cdot \mathbf{p}_{o_K}}_{s_{v_n,c_m,o_K}} + b_{c_m}. \end{aligned} \tag{7}$$

The equation above directly decomposes the prediction value of the target node into orbit units. Each term represents the magnitude of the decomposed prediction value, similarly indicating the contribution of orbits, akin class weight decomposition. Therefore, we define the contribution of orbit $o_k$ to the class $c_m$ prediction of target node $v_n$ as node-class-orbit score, $s_{v_n,c_m,o_k}$. To provide

---

**Algorithm 2** Class-Orbit Score Learning

---

**Input:** A set of orbit bases $\{\mathbf{p}_{o_0}, \cdots, \mathbf{p}_{o_k}, \cdots, \mathbf{p}_{o_{72}}\}$, a set of class weight vectors $\{\mathbf{w}_{c_1}, \cdots, \mathbf{w}_{c_m}, \cdots, \mathbf{w}_{c_{|\mathcal{C}|}}\}$.

**Output:** A set $S$ of selected orbit's class-orbit scores $s_{c_m, o_k}$

**Variables:** A vector composed of an element as a selected orbit's class-orbit score $\mathbf{S}_{c_m}$.

**Initialize** $S$ as an empty set

1: **for** m=1 to $|\mathcal{C}|$ (the number of classes) **do**
2:   Initialize $l_{min} = \infty$
3:   Initialize $\mathbf{P}_{c_m}$ as an empty matrix
4:   **while do**
5:     $selected\_orbit = \arg\min_{o_k \in \mathcal{O}} \min_{\mathbf{S}_{c_m} > 0} \|\mathbf{w}_{c_m} - [\mathbf{P}_{c_m} | \mathbf{p}_{o_k}] \mathbf{S}_{c_m}\|$
6:     $l = \|\mathbf{w}_{c_m} - [\mathbf{P}_{c_m} | \mathbf{p}_{selected\_orbit}] \mathbf{S}_{c_m}\|$
7:     **if** $l < l_{min}$ **then**
8:       $l_{min} = l$
9:       $\mathbf{P}_{c_m} = [\mathbf{P}_{c_m} | \mathbf{p}_{selected\_orbit}]$
10:    **else**
11:      $S.add(\text{all } s_{c_m, o_k} \text{ in the } \mathbf{S}_{c_m})$
12:      **break**
13:    **end if**
14:  **end while**
15: **end for**
16: **Return** $S$

---

instance-level explanations, UO-Explainer extracts the orbit $o^*_{v_n, c_m}$ with the highest node-class-score as follows:

$$o^*_{v_n, c_m} = \arg\max_{o_k \in \mathcal{O}} \left\{ s_{v_n, c_m, o_0}, \cdots, s_{v_n, c_m, o_k}, \cdots, s_{v_n, c_m, o_K} \right\}. \tag{8}$$

Unlike the model-level explanation, which provides the orbit with the highest contribution and its corresponding graphlet as an explanation, instance-level explanations must provide subgraphs around the target node within the input graph. To achieve this, we use the search algorithm based on the Breadth-First Search (BFS). These algorithms initiate the search from the target node and explore neighboring nodes by verifying whether their connectivity matches the highest contributed orbit's corresponding graphlets. A detailed algorithm is shown in Appendix B. Through this search, UO-Explainer can extract a subgraph within the input graph that matches the highest-contributing orbit for the target node, i.e., the explored subgraph is provided as an instance-level explanation along with the target node.

### 3.4 TIME COMPLEXITY ANALYSIS

Training UO-Explainer consists of two main processes: orbit basis learning and class-orbit score learning, with the time complexity for each process detailed below. The time complexity of **orbit basis learning** is $O(|\mathcal{O}||\mathcal{V}|)$, where training is conducted for each orbit basis across all nodes in the input graph, as outlined in Algorithm 1. Here, $|\mathcal{O}|$ represents the number of orbits used for explanation, and $|\mathcal{V}|$ denotes the total number of nodes in the input graph. The time complexity for **class-orbit score learning** depends on the number of orbits selected through greedy search for each class weight. In our experiments, the number of orbits selected did not exceed 5, rendering this complexity component negligible. Consequently, the overall time complexity for this phase is represented as $O(|\mathcal{C}|)$, following the procedure in Algorithm 2, where $\mathcal{C}$ indicates the number of classes. In short, the overall time complexity for training UO-Explainer is $O(|\mathcal{O}||\mathcal{V}| + |\mathcal{C}|) \approx O(|\mathcal{V}|)$.

The time complexities of the baseline methods are as follows: D4Explainer has a time complexity of $O(|\mathcal{V}|^3)$, GNNExplainer is $O(|\mathcal{V}||\mathcal{E}|)$ where $|\mathcal{E}|$ denotes the number of edges, PGExplainer and TAGE have a time complexity of $O(|\mathcal{E}|)$, and MotifExplainer operates with a complexity of $O(|\mathcal{V}||\mathcal{M}|)$, where $|\mathcal{M}|$ represents the number of motifs used in explanations. Therefore, in terms of time complexity, UO-Explainer is less demanding compared to D4Explainer and GNNExplainer. Alongside such analysis, our unified model is capable of providing both model-level and instance-level explanations simultaneously, thus demonstrating its competitiveness in terms of time efficiency. A more detailed time complexity analysis and experiments are described in Appendix C.

## 4 RELATED WORK

GNN explanation methods can be categorized into model-level and instance-level, each with its own scope of explanation (Yuan et al., 2022). Model-level methods (Yuan et al., 2020; Shin et al., 2022; Azzolin et al., 2022) offer explanations that describe the general behavior underlying GNN predictions regarding to specific class. For example, XGNN (Yuan et al., 2020) aims to generate model-level patterns that maximize the predictive probability of a certain class by training a graph generator through reinforcement learning. Similarly, PAGE (Shin et al., 2022) employs graph representation vectors to iteratively search for human-interpretable prototype graphs. Moreover, GLGExplainer (Azzolin et al., 2022) provides general model-level patterns by aggregating instance-level explanations into logical formulas, utilizing an Entropy-Logic Explainer (E-LEN) (Barbiero et al., 2022; Ciravegna et al., 2023). However, these studies primarily focus on graph classification tasks and do not directly apply to node classification tasks. To bridge this gap, D4Explainer (Chen et al., 2023) introduces a method for providing counterfactual model-level explanations for node classification tasks, alongside instance-level explanations, based on a diffusion model. This approach, however, results in a high training cost for the explainer incurred by iterative diffusion- and deonising steps. Moreover, none of the methods for model-level explanations fully consider the user-centric explanation unit, hindering human-interpretability.

Instance-level methods (Yu & Gao, 2022; Ying et al., 2019; Luo et al., 2020; Xie et al., 2022; Vu & Thai, 2020; Wang et al., 2023; Xiong et al., 2023; Zhang et al., 2023; Ye et al., 2024; Lu et al., 2024) provide explanations in the form of subgraph to elucidate the prediction of a specific instance by unveiling relational structures with high contributions in the input graph. Notably, GNNExplainer (Ying et al., 2019) stands as an early instance-level explainer, optimizing edge and feature masks to maximize mutual information with GNN prediction results. PGExplainer (Luo et al., 2020) leverages node representation vectors and trains a parameterized mask predictor to optimize edge masks for explanation in inductive settings. TAGE (Xie et al., 2022) explains the GNN embedding models, allowing efficient explanations for multiple downstream tasks. All these methods entail learning edge masks to present masked graphs as instance-level explanation subgraphs. Despite connectivity constraints, it often fails to generate connected subgraphs, resulting in less intuitive explanations. On the other hand, MotifExplainer (Yu & Gao, 2022) provides an instance-level explanation that does not rely on edge masks. It computes embedding vectors for each motif and extracts explanations by restoring the GNN's prediction value using an attention network. However, it overlooks both the universal importance of specific motifs and the orbit-based isomorphism when extracting motifs. Furthermore, the abovementioned methods are limited to providing explanations in the unified framework capable of simultaneously providing both model-level and instance-level explanations under the human-interpretable units of interest.

## 5 EXPERIMENT

We evaluate explanations provided by UO-Explainer at the model-level and instance-level on synthetic and real-world datasets. These extensive experiments include quantitative and qualitative analysis of UO-Explainer's performance compared to recent baselines. For detailed experimental settings, please refer to Appendix D.

### 5.1 DATASETS AND BASELINES

We conducted experiments using five synthetic datasets and three real-world data sets. Synthetic datasets such as Random Graph (Holme & Kim, 2002), BA-Shapes (Ying et al., 2019), BA-Community, Tree-Cycle, and Tree-Grid are used to evaluate GNN explanation methods to compare the generated explanation based on pre-defined ground truths of each dataset. Also, real-world datasets such as Protein-Protein Interaction (PPI) (Zitnik & Leskovec, 2017), LastFM-Asia (Rozemberczki & Sarkar, 2020), and Gene (Zitnik & Leskovec, 2017) are used for node classification tasks. The detailed information and statistics of each dataset are described in Appendix F.

Among the existing methods, D4Explainer (Chen et al., 2023) and GLGExplainer (Azzolin et al., 2022) are the only existing framework that provides model-level explanations for the node classification task. For instance-level explanations, we set baselines based on common methods that learn edge masks, such as GNNExplainer (Ying et al., 2019), PGExplainer (Luo et al., 2020), TAGE

Table 1: Model-level explanation on random graph datasets: (a) with 2 or 3-layer GCN, and (b) with 2 or 3-layer GIN. Each task is to classify whether the node belongs to the orbit corresponding task number. The evaluation metric is the *Sub-recall*. The best performances are shown in **bold**.

(a)

| Task number | 8 | 11 | 16 | 21 | 27 | 31 | 32 | 33 | 35 | 39 | 45 | 47 | 49 | 57 | 59 | 60 | 61 | 62 | 64 |
|---|---|---|---|---|---|---|---|---|---|---|---|---|---|---|---|---|---|---|---|
| Ground-truth orbit | $o_8$ | $o_{11}$ | $o_{16}$ | $o_{21}$ | $o_{27}$ | $o_{31}$ | $o_{32}$ | $o_{33}$ | $o_{35}$ | $o_{39}$ | $o_{45}$ | $o_{47}$ | $o_{49}$ | $o_{57}$ | $o_{59}$ | $o_{60}$ | $o_{61}$ | $o_{62}$ | $o_{64}$ |
| D4Explainer | 0.2 | 0.8 | 0.4 | 0.4 | 0.2 | 0.4 | 0.4 | 0.4 | **1.0** | 0.8 | 0.2 | 0.4 | 0.4 | 0.8 | 0.2 | 0.0 | 0.6 | **0.6** | 0.2 |
| GLGExplainer | 0.8 | 0.6 | 0.8 | 0.6 | 0.8 | **1.0** | **1.0** | **0.8** | 0.8 | 0.8 | 0.8 | 0.6 | 0.6 | **1.0** | 0.6 | **0.6** | 0.8 | 0.0 | **1.0** |
| **UO-Explainer** | **1.0** | **1.0** | **1.0** | **1.0** | **1.0** | **1.0** | **1.0** | 0.0 | **1.0** | **1.0** | **1.0** | **1.0** | **1.0** | **1.0** | **1.0** | 0.0 | **1.0** | 0.0 | **1.0** |

(b)

| Task number | 8 | 11 | 16 | 21 | 27 | 31 | 32 | 33 | 35 | 39 | 45 | 47 | 49 | 57 | 59 | 60 | 61 | 62 | 64 |
|---|---|---|---|---|---|---|---|---|---|---|---|---|---|---|---|---|---|---|---|
| Ground-truth orbit | $o_8$ | $o_{11}$ | $o_{16}$ | $o_{21}$ | $o_{27}$ | $o_{31}$ | $o_{32}$ | $o_{33}$ | $o_{35}$ | $o_{39}$ | $o_{62}$ | $o_{47}$ | $o_{49}$ | $o_{57}$ | $o_{59}$ | $o_{60}$ | $o_{61}$ | $o_{45}$ | $o_{64}$ |
| D4Explainer | 0.8 | 0.6 | 0.6 | **1.0** | 0.6 | 0.8 | 0.8 | 0.6 | **1.0** | **1.0** | 0.6 | 0.4 | 0.8 | **1.0** | 0.8 | 0.8 | 0.2 | 0.4 | 0.6 |
| GLGExplainer | 0.8 | 0.8 | **1.0** | 0.4 | **1.0** | **1.0** | **1.0** | **1.0** | 0.8 | **1.0** | **1.0** | 0.6 | 0.8 | **1.0** | 0.6 | 0.8 | **1.0** | 0.0 | **1.0** |
| **UO-Explainer** | **1.0** | **1.0** | **1.0** | **1.0** | **1.0** | **1.0** | **1.0** | **1.0** | **1.0** | **1.0** | **1.0** | **1.0** | **1.0** | **1.0** | **1.0** | **1.0** | **1.0** | **1.0** | **1.0** |

Table 2: Model-level explanation results on synthetic datasets. The evaluation metric is the Sub-recall.

| | BA-Shapes | | | BA-Community | | | | | | Tree-Grid | | | Tree-Cycle |
|---|---|---|---|---|---|---|---|---|---|---|---|---|---|
| | class1 | class2 | class3 | class1 | class2 | class3 | class5 | class6 | class7 | class1 | class2 | class3 | class1 |
| Ground-truth Orbit | $o_{58}$ | $o_{57}$ | $o_{56}$ | $o_{58}$ | $o_{57}$ | $o_{56}$ | $o_{58}$ | $o_{57}$ | $o_{56}$ | $o_{73}$ | $o_{74}$ | $o_{75}$ | $o_{76}$ |
| D4Explainer | 0.4 | 0.6 | 0.4 | 0.8 | 0.2 | 0.8 | 0.0 | 0.6 | 0.4 | 0.6 | 0.0 | 0.2 | 0.8 |
| GLGExplainer | **1.0** | **1.0** | **1.0** | **1.0** | 0.8 | 0.2 | 0.8 | **1.0** | 0.8 | 0.0 | 0.8 | 0.2 | **1.0** |
| **UO-Explainer** | **1.0** | **1.0** | **1.0** | **1.0** | **1.0** | **1.0** | **0.8** | **1.0** | **1.0** | **0.8** | **1.0** | **1.0** | **1.0** |

(Xie et al., 2022), MixupExplainer (Zhang et al., 2023), SAME (Ye et al., 2024), and EIG (Lu et al., 2024). Additionally, MotifExplainer (Yu & Gao, 2022), which provides explanations based on motifs similar to our model, was also set as a baseline.

## 5.2 EVALUATION METRIC

We evaluate the quality of explanations using Sparsity, Fidelity, Edge-recall, and Sub-recall as evaluation metrics. *Sparsity* (Li et al., 2022) refers to the ratio of edges in the explanation compared to the total number of edges in the computation graph of the target node. High sparsity implies that the proposed explanation has a small number of edges. *Fidelity* (Li et al., 2022) measures the difference in the probability values when the explanation is excluded from the computation graph based on the target node. *Edge-recall* indicates how many edges in the explanations match the edges in the ground truths. *Sub-Recall* indicates the proportion of correct answers that the entire presented explanations match with ground truths. Notably, equations of *Sparsity* and *Fidelity* are described in the Appendix D.4. Additionally, we conducted experiments five times and then reported the average and standard deviations in Appendix E.

## 5.3 RESULTS: MODEL-LEVEL EXPLANATIONS

We first validate whether the UO-Explainer can identify the correct orbits for model-level explanations. We pre-train 2 or 3-layer GCN models (Kipf & Welling, 2017) and 2 or 3-layer GIN models (Xu et al., 2019) on Random Graph datasets. We pre-train 2 or 3-layer GCN models (Kipf & Welling, 2017) and 2 or 3-layer GIN models (Xu et al., 2019) on Random Graph datasets. Each task is to classify whether the node belongs to the orbit corresponding task number. Consequently, explanation methods are expected to provide the ground-truth pattern in the form of the orbit (target node) with its corresponding graphlet (pattern) for each task. Tasks with accuracy below 0.8 are excluded as they are unlikely to yield accurate explanations. We use the *Sub-recall* metric to evaluate whether the provided explanation matches the ground-truth pattern.

In Table 1, UO-Explainer shows superior performance compared to other baselines. In particular, the UO-Explainer constantly provides model-level explanations matching to ground truths for all tasks except 33, 60, and 62 in the GCN model as shown in (a). This limitation may arise from the GNN's expressiveness, failing to learn intended orbits during the pre-training. To address this, we conducted experiments in the same manner using GIN, known for better expressiveness. The results are shown in (b) of Table 1 that UO-Explainer accurately provides the explanations matching the

Table 3: Instance-level explanation results on synthetic datasets. The best performances on each dataset are shown in **bold**.

| | BA-Shapes | | | BA-Community | | | Tree-Grid | | | Tree-Cycle | | |
|---|---|---|---|---|---|---|---|---|---|---|---|---|
| | Sub-recall | Edge-recall | Fidelity | Sub-recall | Edge-recall | Fidelity | Sub-recall | Edge-recall | Fidelity | Sub-recall | Edge-recall | Fidelity |
| GNNExplainer | 0.004 | 0.616 | 0.580 | 0.006 | 0.491 | 0.653 | 0.000 | 0.629 | 0.872 | 0.119 | 0.699 | 0.724 |
| PGExplainer | 0.760 | 0.915 | 0.574 | 0.238 | 0.667 | 0.652 | 0.000 | 0.647 | 0.876 | 0.926 | 0.992 | 0.732 |
| TAGE | 0.682 | 0.900 | 0.601 | 0.352 | 0.754 | 0.672 | 0.003 | 0.693 | 0.874 | 0.963 | 0.994 | 0.734 |
| MixupExplainer | 0.696 | 0.906 | 0.612 | 0.496 | 0.857 | 0.693 | 0.047 | 0.712 | 0.8770 | 0.930 | 0.994 | 0.734 |
| SAME | 0.343 | 0.720 | 0.547 | 0.132 | 0.680 | 0.642 | 0.000 | 0.238 | 0.846 | 0.101 | 0.635 | 0.692 |
| EiG-Search | 0.878 | 0.520 | 0.605 | 0.078 | 0.681 | 0.695 | 0.004 | 0.723 | 0.885 | 0.083 | 0.812 | 0.703 |
| MotifExplainer | 0.873 | 0.890 | 0.548 | 0.423 | 0.714 | 0.683 | 0.793 | 0.857 | 0.879 | 0.991 | 0.993 | 0.736 |
| **UO-Explainer** | **0.948** | **0.984** | **0.623** | **0.921** | **0.970** | **0.716** | **0.859** | **0.900** | **0.888** | **1.000** | **1.000** | **0.737** |

Table 4: Instance-level explanation results on real datasets. The best fidelity on each dataset is shown in **bold**. ⋆ notation indicates the lower sparsity setting.

| | PPI | | | | | | | | | | | | LastFM Asia | |
|---|---|---|---|---|---|---|---|---|---|---|---|---|---|---|
| | Task0 | | Task1 | | Task2 | | Task3 | | Task4 | | Task5 | | | |
| | Fidelity | Sparsity | Fidelity | Sparsity | Fidelity | Sparsity | Fidelity | Sparsity | Fidelity | Sparsity | Fidelity | Sparsity | Fidelity | Sparsity |
| GNNExplainer | 0.100 | 0.973 | 0.325 | 0.999 | 0.417 | 0.999 | 0.480 | 0.999 | 0.250 | 0.999 | 0.331 | 0.999 | 0.114 | 0.974 |
| PGExplainer* | 0.023 | 0.973 | 0.155 | 0.999 | 0.223 | 0.999 | 0.101 | 0.999 | 0.200 | 0.999 | 0.005 | 0.999 | 0.011 | 0.974 |
| TAGE* | 0.031 | 0.973 | 0.109 | 0.999 | 0.233 | 0.999 | 0.138 | 0.999 | 0.214 | 0.999 | 0.101 | 0.999 | 0.086 | 0.974 |
| MixupExplainer | 0.005 | 0.973 | 0.002 | 0.999 | 0.257 | 0.999 | 0.246 | 0.999 | 0.246 | 0.999 | 0.129 | 0.999 | 0.100 | 0.974 |
| EiG-Search | 0.180 | 0.973 | 0.269 | 0.999 | 0.180 | 0.999 | 0.379 | 0.999 | 0.100 | 0.999 | 0.180 | 0.999 | 0.095 | 0.974 |
| SAME | 0.022 | 0.973 | 0.189 | 0.999 | 0.194 | 0.999 | 0.189 | 0.999 | 0.034 | 0.999 | 0.129 | 0.999 | 0.049 | 0.974 |
| MotifExplainer | 0.070 | 0.992 | 0.074 | 0.999 | 0.012 | 0.999 | 0.129 | 0.999 | 0.097 | 0.999 | 0.050 | 0.999 | 0.085 | 0.994 |
| **UO-Explainer** | **0.423** | 0.999 | **0.358** | 0.999 | **0.425** | 0.999 | **0.510** | 0.999 | **0.623** | 0.999 | **0.413** | 0.999 | **0.115** | 0.993 |

correct ground-truth for all tasks. These findings confirm that UO-Explainer is capable of detecting various orbits and providing correct explanations while UO-Explainer's explanations mirror the expressiveness of the GNN's embedding model. Notably, comparing the performance of original GNNs to the decomposed class weights model, the performance degradation is less than 5%. On the other hand, D4Explainer and GLGExplainer show slightly improved or even decreased performance in the GIN setting and fail to provide consistent explanations matched with the ground truth.

As observed in Table 2, UO-Explainer also outperforms other baselines on the BA-Shapes and BA-Community dataset. Using 3-layer GCNs for this experiment, UO-Explainer successfully provides accurate model-level explanations for each class on both datasets. Specifically on the BA-Shapes dataset, we observe the explanations that the house-like motif plays a crucial role in node classification by detecting the orbit such as $o_{56}, o_{57}$, and $o_{58}$ in the graphlet $g_{23}$ as explanations. This explanation sheds light on the overall behavior of the GNN beyond individual nodes, enabling a broader interpretation of GNNs. Moreover, our approach provides orbit-corresponding graphlets, allowing us to determine the topological positions of nodes within the motif for each class. Thus, UO-Explainer shows that the GNN recognizes the house-like motif as an important pattern for node classification, assigning nodes on the roof, floor, and top of the roof of the house-like motif to classes 1, 2, and 3, respectively. On the BA-Community dataset, UO-Explainer also finds the ground-truth pattern as the model-level explanation by detecting the orbit such as $o_{56}, o_{57}$, and $o_{58}$ in the graphlet $g_{23}$ as explanations, since the dataset is a union of two BA-SHAPES graphs. In conclusion, the experimental result demonstrates that orbits as pre-defined explanation units of UO-Explainer serve crucial patterns of the specific classes for prediction, showing accurate and consistent explanations.

## 5.4 Results: Instance-level Explanations

Except for MotifExplainer, all baselines provide explanations in the form of subgraphs obtained by extracting edges exceeding specific threshold values or ranking the top-$k$ edge considered important. For fair experiments, explanations composed of top-$k$ edges were extracted considering a sparsity level similar to that of UO-Explainer. In the case of MotifExplainer, we extract one motif as an instance-level explanation. UO-Explainer used a subgraph within the input graph that matches the highest-contributing orbit for the target node for the instance-level explanation as mentioned in Section 3.3.

The experimental results on synthetic datasets are shown in Table 3. UO-Explainer outperforms the baseline methods across all evaluation metrics while maintaining a comparable sparsity level. Notably, UO-Explainer achieves higher sub-recall, indicating accurate detection of ground-truth subgraphs as explanations. In cases such as Tree-grid and Tree-cycle, where grid- and cycle-shaped graphlets do not exist within the 2- to 5-node graphlets, we employ grid and cycle graphlets along with their corresponding orbits and include them as units of explanation. We note that by defining custom graphlets based on background knowledge or extracted rules tailored to specific problem

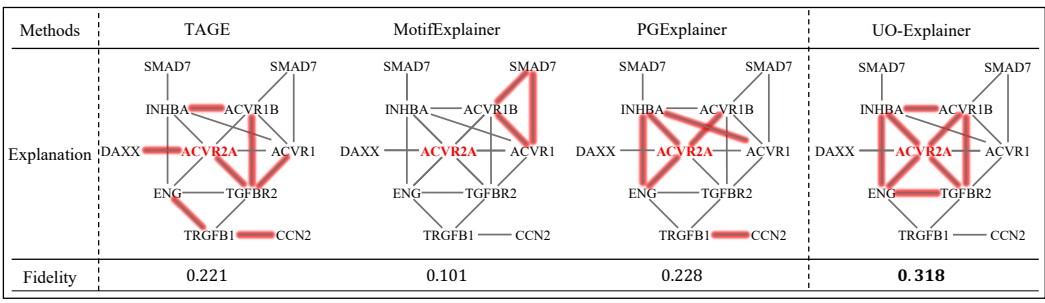

Figure 4: Visualization of the explanations provided in the Gene dataset. Each node represents the ID of a gene, and the red nodes correspond to the target gene mentioned in the explanation. The red edges denote the edges included in the subgraph provided as part of the explanation.

settings, the proposed method can be extended beyond the pre-defined 2- to 5-node graphlets and orbits.

Table 4 shows the performance on real datasets, where UO-Explainer outperforms other baselines in the fidelity metric, except for task 0 of the PPI dataset. MotifExplainer struggles due to extracting too many motifs, leading to less meaningful explanations. PGExplainer and TAGE also show poor performance, often identifying the same or irrelevant edges across nodes, regardless of experimental changes. GNNExplainer, trained iteratively for node-specific explanations, extracts more relevant edges, especially on larger datasets. The ∗ notation in Table 4 highlights that lower sparsity is needed for competing methods to match UO-Explainer, yet they still include noise edges or less important subgraphs. In contrast, UO-Explainer performs consistently well even under high sparsity, using just one orbit-based subgraph for explanations, demonstrating its ability to provide high-quality instance-level explanations.

## 5.5 CASE STUDY ON GENE DATASET

As the qualitative analysis, we visualized the explanatory subgraphs described in our method and the baselines on the gene dataset. The visualization results are presented in Figure 4. The experiment results demonstrate that PGExplainer and TAGE provide scattered subgraphs with discontinuous edges while the sparsity remains at **0.900** for fair comparison. In contrast, UO-Explainer offers a connected subgraph that is more intuitive while maintaining relatively high fidelity. MotifExplainer also presents a connected subgraph as an explanation but with relatively lower fidelity. Additionally, several studies provide evidence that the genes (TGFBR2 (Massagué & Gomis, 2006), ENG (Breen et al., 2013), INHBA, and ACVR2B (Attisano & Wrana, 2013)) identified by UO-Explainer in the explanations have an impact on the surface receptor signaling pathway, which is the label of the dataset. For example, in (Bottino et al., 2021), it was mentioned that TGFBR2 is one of the TGF-beta receptors that transmit signals within natural killer cells, exerting a significant influence on cell development and the function of natural killer cells. These results imply that UO-Explainer provides human-interpretable explanations compared to other baselines regarding the perspective of interest.

## 6 DISCUSSION AND CONCLUSION

We introduce UO-Explainer, a human-interpretable explanation method that leverages pre-defined units, as requested by users, in a unified framework for node classification models. By utilizing orbits as explanatory units, UO-Explainer decomposes model weights into orbit components, which serve as essential, human-interpretable units within graph domains. Experimental results on both synthetic and real-world datasets demonstrate the effectiveness of UO-Explainer, outperforming baseline methods and delivering higher-quality explanations. UO-Explainer is particularly valuable in scientific applications, such as drug development and education, where domain knowledge is critical. By using pre-defined explanation units, users can uncover meaningful patterns and gain deeper insights through the explanation method.

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

## A    THE FULL SET OF GRAPHLETS AND ORBITS

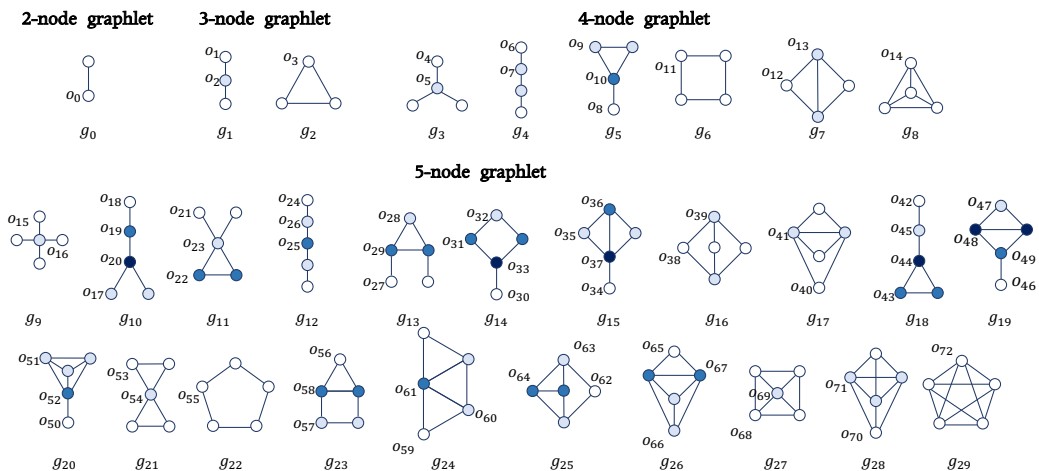

Figure 5: Entire graphlets having 2-5 nodes and orbits. The same color nodes within each graphlet belong to the same orbit.

In this work, we have defined the explanatory units as 0-72 orbits and their corresponding graphlet ranging from 0 to 29 consisting of 2 to 5 nodes. The full set of utilized graphlets and orbits can be observed in Figure 5 (Pržulj et al., 2004; Pržulj, 2007).

## B    ORBIT SEARCH ALGORITHM

In Section 3.1, we have mentioned the implementation of a graphlet search algorithm based on the Breadth-First Search (BFS) algorithm for extracting the orbit with the highest score and its corresponding graphlet on the target node. These algorithms initiate the search from the target node and explore neighboring nodes by verifying whether their connectivity matches the desired graphlets. We present two algorithmic examples for identifying specific orbits in our study. Algorithm 3 describes the process of identifying the $o_{10}$ with its corresponding graphlet $g_5$, while Algorithm 4 outlines the steps for identifying the $o_{56}$ with its corresponding graphlet $g_{23}$. In both algorithms, $N(v)$ denotes the neighbor function to construct the set of neighboring nodes around a given node $v$. Additionally, $C(\cdot, n)$ denotes a combination function to generate the set of combinations consisting of $n$ elements from a given set $\cdot$.

---

**Algorithm 3** Orbit $o_{10}$ Search Algorithm

---

**Input:** A graph $\mathcal{G} = (\mathcal{V}, \mathcal{E})$ where $\mathcal{V} = \{v_1, \cdots, v_i, \cdots, v_{|\mathcal{V}|}\}$ denotes a set of node and $\mathcal{E} = \{(v_i, v_j), \cdots, (v_k, v_l)\}$ denotes a set of edges, Target node $v_n$.
**Output:** A set $L$ which contains edge sets of $g_5$ graphlets that make target node belong to orbit $o_{10}$.
**Initialize**  $L$ as an empty set
1: Neighbor set of the target node $\mathcal{N}_{v_n} = N(v_n)$ where $N$ denotes the neighbor function.
2: A combination set with three elements $\mathcal{C}_1 = C(\mathcal{N}_{v_n}, 3)$ where $C$ denotes Combination function
3: **for** $v_a^{'}, v_b^{'}, v_c^{'}$ in $\mathcal{C}_1$ **do**
4:     a set of candidate combinations $\mathcal{C}_1^{'} = \{(v_a^{'}, v_b^{'}, v_c^{'}), (v_b^{'}, v_a^{'}, v_c^{'}), (v_c^{'}, v_a^{'}, v_b^{'})\}$
5:     **for** $v_a, v_b, v_c$ in $\mathcal{C}_1^{'}$ **do**
6:         **if** $(v_b, v_c) \in \mathcal{E}$ and $\{(v_a, v_b), (v_a, v_c)\} \notin \mathcal{E}$ **then**
7:             $L.add(\{(v_n, v_a), (v_n, v_b), (v_n, v_c), (v_b, v_c)\})$
8:         **end if**
9:     **end for**
10: **end for**
11: **Return** $L$

---

We present a detailed line-by-line explanation of Algorithm 3 as an exemplary demonstration of the orbit search process. This algorithm is designed to find $g_5$ graphlets in which the target node belongs to orbit $o_{10}$. In line 1, we generate the neighbor set $\mathcal{N}_{v_n}$ from the target node $v_n$. Since the number of neighboring nodes within orbit $o_{10}$ in graphlet $g_5$ is three, line 2 constructs the set

$\mathcal{C}_1$ by forming combinations of three nodes from the neighbor set. Within graphlet $g_5$, there is one orbit, $o_8$, that distinguishes it from the two neighboring nodes, $o_9$. Therefore, to ensure that each node combination in $\mathcal{C}_1$ also distinguishes between $o_{10}$ and $o_9$, line 4 generates the set $\mathcal{C}_1'$, which reflects requirement. Notably, the first nodes of each tuple in the set are distinct from the remaining nodes. By observing the connectivity of the nodes in lines 5-6, if it matches the connectivity of the orbits within $g_5$, the edge set of $g_5$ can be obtained in line 7. It is added to the set $L$. This algorithm iterates by considering all possible combinations of neighboring nodes surrounding the target node $v_n$. Therefore, all existing graphlet $g_5$ that make the target node belong to orbit $o_{10}$ are in the output set $L$. This format can be applied to find entire 2-5 node graphlets. However, it is computationally expensive. To alleviate this computational cost, by introducing randomness during the generation of the neighbor set (line 1) or combination set in the algorithm(line 2 and line 4), it becomes possible to sample the desired number of graphlets. The performance as the number of samples is discussed in Table 7.

---

**Algorithm 4** Orbit $o_{56}$ Search Algorithm

---

**Input:** A graph $\mathcal{G} = (\mathcal{V}, \mathcal{E})$ where $\mathcal{V} = \{v_1, \cdots, v_i, \cdots, v_{|\mathcal{V}|}\}$ denotes a set of nodes and $\mathcal{E} = \{(v_i, v_j), \cdots, (v_k, v_l)\}$ denotes a set of edges, Target node $v_n$.
**Output:** A set $L$ which contains edge sets of $g_{23}$ graphlets that make target node belong to orbit $o_{56}$.
**Initialize** $L$ as an empty set
1: Neighbor set of the target node $\mathcal{N}_{v_n} = N(v_n)$ where $N$ denotes the neighbor function.
2: A combination set with three elements $\mathcal{C}_1 = C(\mathcal{N}_{v_n}, 2)$ where $C$ denotes Combination function
3: **for** $v_a, v_b$ in $\mathcal{C}_1$ **do**
4:     **if** $v_b, v_c \in \mathcal{E}$ **then**
5:         Neighbor set of $v_a$, $\mathcal{N}_{v_a} = N(v_a)$
6:         Neighbor set of $v_b$, $\mathcal{N}_{v_b} = N(v_b)$
7:         **for** $v_c$ in $\mathcal{N}_{v_a}$ **do**
8:             **for** $v_d$ in $\mathcal{N}_{v_b}$ **do**
9:                 **if** $(v_c, v_d) \in \mathcal{E}$ and $\{(v_n, v_c), (v_n, v_d), (v_b, v_c), (v_a, v_d)\} \notin \mathcal{E}$ **then**
10:                     $L.add(\{(v_n, v_a), (v_n, v_b), (v_a, v_b), (v_a, v_c), (v_b, v_d), (v_c, v_d)\})$
11:                 **end if**
12:             **end for**
13:         **end for**
14:     **end if**
15: **end for**
16: **Return** $L$

---

## C   Time Complexity Analysis

When UO-Explainer provides instance-level explanations, there are four time-consuming processes: 1) Pre-processing for finding the existence of the orbit, 2) Orbit basis learning, 3) Class-orbit score learning (Model-level explanation generation), and 4) Generating instance-level explanations.

**1) Pre-processing for finding the existence of each orbit.** As mentioned in Section 3.1, to learn the orbit basis, we must find the existence of each orbit for every node, as represented by the equation 1. This pre-processing can be conducted to search 2-5 node graphlets for each node. The time complexity of finding each graphlet that includes specific orbits, as can be inferred from Algorithm 3 and Algorithm 4, is $O(d^{k-1})$[1]. Here, $d$ represents the maximum node degree of the graph data, and $k$ denotes the number of nodes included in each graphlet and is less than 5. Therefore, the time complexity of pre-processing for finding all 2-5 node graphlets that include a total 0-72 orbit for an entire node of graph data is $O(|\mathcal{O}||\mathcal{V}|d^{k-1})$ where $|\mathcal{O}|$ denotes the number of edges and $|\mathcal{V}|$ represents the number of nodes.

**2) Orbit basis learning.** The time complexity of orbit basis learning is $O(|\mathcal{O}||\mathcal{V}|)$ since training is performed for each orbit basis over all nodes as shown in Algorithm 1.

**3) Class-orbit score learning.** Class-orbit score is trained based on the number of orbits selected by the greedy search for each class weight. In our experiments, the number of selected orbits was less than or equal to 5, so this can be sufficiently neglected. Therefore, the time complexity of class-orbit score learning considers only the number or class $\mathcal{C}$; $O(|\mathcal{C}|)$ as shown in Algorithm 2. The training time required for UO-Explainer and the baselines can be found in Table 7. Model-level explanations

---

[1]As referenced from N. K. Ahmed, J. Neville, R. A. Rossi, and N. Duffield. Efficient Graphlet Counting for Large Networks. *In Proceedings of the International Conference on Data Mining*, 2015, pp. 1-10.

Table 5: The training time required for the baselines and UO-Explainer for explanation in the gene dataset. The training time of the explainer is significantly influenced by the epochs. Therefore, we set the epochs for each method through a hyper-parameter search to achieve the highest fidelity.

| | UO-Explainer | GNNExplainer | MotifExplainer | PGExplainer | TAGE |
|---|---|---|---|---|---|
| Training Time(s) | 268 (orbit basis learning)+21 (class-orbit score learning) | $1,681$ | $6,074$ | 98 | 204 |

Table 6: The training time required for the baselines and UO-Explainer for explanation in the gene dataset. The training time of the explainer is significantly influenced by the epochs. Therefore, we set the epochs for each method through a hyper-parameter search to achieve the highest fidelity.

| | Instance and Model-level | Instance-level | | | | Model-level |
|---|---|---|---|---|---|---|
| | UO-Explainer | GNNExplainer | MotifExplainer | PGExplainer | TAGE | D4Explainer |
| Training Time(s) | 268 (orbit basis learning)+21 (class-orbit score learning) | $1,681$ | $6,074$ | 98 | 204 | 502 |

can be provided immediately after class-orbit score learning. Therefore, the training time of the UO-Explainer listed in Table 6 equals the time required to provide model-level explanations.

**4) Generating instance-level explanations.** To provide instance-level explanations, it is necessary to search graphlets that include the given orbits as explanations for each node and select just one graphlet having the highest fidelity. Therefore, the time complexity is $O(|\mathcal{V}| d^{k-1})$ since graphlets need to be directly found for each node. However, instead of finding all graphlets around each node, we sample a predetermined number of graphlets. Thus, the actual computational cost can be significantly reduced compared to $O(|\mathcal{V}| d^{k-1})$, while preserving high explanation performance. The required time and fidelity based on the number of sampled graphlets are shown in Table 7. Even with a time difference of more than 7 times, as observed when comparing the cases of not sampling and sampling 70 graphlets for explanation extraction, it can be observed that the fidelity values do not significantly decrease.

# D  DETAILED EXPERIMENTAL SETTINGS

In this section, we cover the details of experiments that were not discussed in the main text, such as the structure of the pre-trained GNN and the hyper-parameter settings of the UO-Explainer.

## D.1  DETAILS OF PRE-TRAINED GNNS

We provide detailed information about pre-trained GNNs for datasets other than the random graph, as already described in Section 5.3. For each dataset, we utilized a pre-trained GNN architecture consisting of a 3-layer GCN as the embedding model and a 1-layer MLP as the downstream model. The hyper-parameters and split ratio are shown in Table 8.

## D.2  HYPER-PARAMETER SETTINGS FOR UO-EXPLAINER

UO-Explainer has a total of five hyper-parameters including batch size, epochs, learning rate (LR) for orbit basis learning, and additional epochs, learning rate (LR) for class-orbit score learning. Hyper-parameters used in the experiments are reported in Table 9.

## D.3  DETAILS OF BASELINES

In our code implementation, we primarily utilized the PyTorch (Paszke et al., 2019) and PyTorch Geometric (Fey & Lenssen, 2019) frameworks. Additionally, for the GNNExplainer and PGExplainer, we employed implementations from the PyTorch Geometric framework. For the TAGE, we employed implementations from the Dive into Graph (DIG) (Liu et al., 2021) framework. For MotifExplainer, we utilized the code provided in the supplementary material available on OpenReview.net, which was suitably modified following advice received in correspondence with the authors. This revised implementation is incorporated within our publicly accessible code.

Furthermore, to ensure the equity of our experiments, we diligently endeavored to explore the hyper-parameter space of the baselines as extensively as possible. The explored hyper-parameter

Table 7: The time required and fidelity based on the number of sampled graphlets of UO-Explainerin the gene dataset. We conducted five experiments for each method and reported the mean and standard deviation values. The "Entire" refers to the extraction of all graphlets surrounding the target node without sampling.

| # of sampled graphlets | 1 | 10 | 30 | 50 | 70 | 100 | Entire |
|---|---|---|---|---|---|---|---|
| Time(s) | $14.572 \pm 0.208$ | $136.342 \pm 2.305$ | $379.342 \pm 22.438$ | $597.283 \pm 3.865$ | $838.991 \pm 3.587$ | $1183.459 \pm 5.913$ | $6321.783 \pm 8.462$ |
| Fildelity | $0.157 \pm 0.018$ | $0.256 \pm 0.009$ | $0.297 \pm 0.011$ | $0.309 \pm 0.014$ | $0.326 \pm 0.008$ | $0.324 \pm 0.007$ | $0.3543 \pm 0$ |

Table 8: Details of pre-trained GNNs. LR means learning rate.

| Dataset | BA-Sahpes | BA-Community | PPI0 | PPI1 | PPI2 | PPI3 | PPI4 | PPI5 | LastFM Asia | Gen | Tree-Cycle | Tree-Grid |
|---|---|---|---|---|---|---|---|---|---|---|---|---|
| LR | 0.001 | 0.001 | 0.003 | | | | | | 0.001 | 0.001 | 0.001 | 0.001 |
| Epoch | 2,000 | 5,000 | 300 | | | | | | 600 | 100 | 600 | 3,000 |
| Hidden Dimension | 10 | 30 | 200 | | | | | | 30 | 100 | 30 | 30 |
| Train:Val:Test | 8:1:1 | 8:1:1 | 10:1:1 | | | | | | 8:1:1 | 8:1:1 | 8:1:1 | 10:1:1 |
| Train Accuracy | 0.982 | 0.998 | 9.998 | 0.998 | 0.997 | 0.998 | 0.994 | 0.998 | 0.997 | 0.999 | 0.993 | 0.987 |
| Val Accuracy | 0.943 | 0.723 | 0.953 | 0.959 | 0.970 | 0.962 | 0.976 | 0.981 | 0.814 | 0.744 | 0.966 | 0.878 |
| Test Accuracy | 0.971 | 0.800 | 0.971 | 0.967 | 0.973 | 0.957 | 0.980 | 0.936 | 0.841 | 0.686 | 0.966 | 0.863 |

space for GNNExplainer includes (lr, epoch, edge size, node feature size, edge entropy, and node feature entropy). For each dataset in this study, the experiment was conducted by selecting the combination of hyper-parameters (100-300-600, 0.01-0.05-0.1, 0.005-0.01-0.1, 0.05-0.1, 0.5-1.0, 0.5-1.0) that yielded the highest performance. The explored hyper-parameter space for PGExplainer includes (lr, epoch, edge size, edge entropy, and dimension of MLP). For each dataset in this study, the experiment was conducted by selecting the combination of hyper-parameters (0.001-0.003-0.01-0.5-0.1, 0.01-0.05-0.1, 0.05-0.1, 64-128) that yielded the highest performance. The explored hyper-parameter space for TAGE includes (lr, epoch, batch size coefficient size, coefficient entropy, and loss type). For each dataset in this study, the experiment was conducted by selecting the combination of hyper-parameters (0.000005-0.00005-0.0005-0.005-0.05-0.1, 1-3-5-10-20, 4-16-64-128, 0.01-0.05, 0.0005-0.005, 'NCE'-'JSE') that yielded the highest performance. The explored hyper-parameter space for MotifExplainer includes (lr, epoch, embedding dimension of attention module). For each dataset in this study, the experiment was conducted by selecting the combination of hyper-parameters (0.001-0.005-0.01, 30-50-100) that yielded the highest performance. Furthermore, the motif extraction rules for MotifExplainer encompass three categories: 1) cycles within the graph, 2) edges excluding cycles, and 3) motifs constituted by the fusion of cycles involving two or more identical nodes. In our experiments, when the number of motifs extracted from the graphs was fewer than 5000, all three rules were employed to conduct the experiments. However, in cases where the number exceeded 5000 motifs, a random sample of 5000 motifs was utilized.

## D.4 EVALUATION METRICS

Sparsity (Li et al., 2022) means the proportion of the presented explanation in the computation graph based on the target node as follows:

$$Sparsity = \frac{1}{|\mathcal{V}|} \sum_{i=1}^{|\mathcal{V}|} 1 - \frac{\left|\mathcal{G}_{v_i}^{ex}\right|}{\left|\mathcal{G}_{v_i}\right|} \tag{9}$$

where $\mathcal{G}_{v_i}^{ex}$ denotes the subgraph presented as the explanation for the node $v_i$, $\mathcal{G}_{v_i}$ denotes to the computation graph, and $|\mathcal{G}|$ means the number of edges in the graph $\mathcal{G}$.

Fidelity (Li et al., 2022) is calculated by taking the difference in the probability values of the input graph and the probability values when the explanation is excluded from the computation graph based on the target node, as follows:

$$Fidelity = \frac{1}{|\mathcal{V}|} \sum_{i=1}^{|\mathcal{V}|} f_{prob}(\mathcal{G}_{v_i}) - f_{prob}(\mathcal{G}_{v_i} - \mathcal{G}_{v_i}^{ex}) \tag{10}$$

where $f_{prob}(\mathcal{G}_{v_i})$ refers to the probability values of each node $v_i$ from the trained GNN $f(G)$ with respect to the correct class.

Table 9: Hyper-parameter settings for UO-Explainer. LR means learning rate.

| Dataset | Random Graph | BA-Shapes | BA-Community | PPI | LastFM-Asia | Gene | Tree-Cycle | Tree-Grid |
|---|---|---|---|---|---|---|---|---|
| Epochs (Orbit basis learning) | 3,000 | 3,000 | 3,000 | 3,000 | 1,000 | 1,000 | 1,000 | 1,000 |
| LR (Orbit basis learning) | 0.005 | 0.005 | 0.005 | 0.001 | 0.003 | 0.005 | 0.005 | 0.005 |
| Batch size (Orbit basis learning) | 256 | 256 | 256 | 256 | 2048 | 256 | 256 | 256 |
| Epochs (Class-orbit score learning) | 2,000 | 1,000 | 1,000 | 1,000 | 1,000 | 1,000 | 1,000 | 1,000 |
| LR (Class-orbit score learning) | 0.003 | 0.005 | 0.005 | 0.005 | 0.005 | 0.005 | 0.005 | 0.005 |

Table 10: Statistics of the dataset used for the experiment. # symbols mean the number.

| Dataset | Random Graph | BA-Shapes | BA-Community | PPI | LastFM-Asia | Gene | Tree-Cycle | Tree-Grid |
|---|---|---|---|---|---|---|---|---|
| Avg # of nodes | 340 | 700 | 1,400 | 2,373 | 7,624 | 857 | 871 | 1,231 |
| Avg # of edges | 2,690 | 4,110 | 8,920 | 66,136 | 55,612 | 13,992 | 1,950 | 3,410 |
| # of classes | 2 | 4 | 8 | 2 | 18 | 2 | 2 | 4 |
| # of tasks | 73 | 1 | 1 | 121 | 1 | 1 | 1 | 1 |

## E  EXPERIMENTAL RESULTS WITH STANDARD DEVIATION.

We conducted five experiments for each method by recording the mean and standard deviation in Table 11, 12, 13, and 14.

## F  DATASETS

We conducted experiments using five synthetic datasets and three real-world datasets. The statistics of datasets are shown in Table 10. Additionally, the detailed information on each dataset is described below.

**Random Graph** (Holme & Kim, 2002): This dataset does not have node feature values, so we performed node classification tasks using only the structural information by setting all node feature values to the same constant value. We set the class of each node based on the existence of its orbit same as each task number. That is, each node has two classes, $c_0$ (orbit doesn't exist) and $c_1$ (orbit exists) for a total of 73 independent tasks, one for each orbit. The ground truth for class $c_1$ is the target orbit $o_k$ for each task.

**BA-Shapes** (Ying et al., 2019): The node classification dataset consists of a BA graph with 300 nodes as the base and 80 "house-like motifs" randomly attached, each consisting of 5 nodes. The house-like motif precisely matches graphlet $g_{23}$. The class is determined by the positions of the nodes within the house-like motif; these node positions precisely match the orbits $o_{58}, o_{57}, o_{56}$ in $g_{23}$. Therefore, the ground truth for each class is set to $o_{58}, o_{57}, o_{56}$.

**BA-Community** (Ying et al., 2019): The node classification dataset is generated by randomly combining two house-like motifs through edges. The first motif is assigned classes $c_0, c_1, c_2, c_3$ based on the positions of the nodes, and the second motif is assigned $c_4, c_5, c_6, c_7$, resulting in a total of eight class labels. The ground truth of each motif is the same as that of the BA-Shapes dataset.

**Tree-Cycle** (Ying et al., 2019): The dataset consists of a balanced binary tree as the base and 80 "six-node cycle motifs" randomly attached. The task is to classify nodes as either not belonging ($c_0$) to or belonging to a circle ($c_1$). The ground truth for $c_1$ class corresponds to the circle motif. However, 2-5 node graphlets do not encompass the six-node circle motif. Thus, we conducted experiments by incorporating the circle motif and the orbit within the circle into candidate graphlets and orbits.

**Tree-Grid** (Ying et al., 2019): The base is identical to that of the Tree-Cycle, with a modification involving the attachment of a "nine-node grid" in place of the circle. The task has been further challengingly augmented to classify each node as belonging to the center ($c_1$), periphery ($c_2$), or cross-lines ($c_3$) based on their positions within the grid. Similar to the case of Tree-Cycle, 2-5 node graphlets do not contain the nine-node grid. Therefore, we included the grid motif and the orbits within the grid in the candidate set for experimentation.

**Protein-Protein Interaction (PPI)** (Zitnik & Leskovec, 2017): This is a node classification dataset that represents protein-protein interactions in various human tissues using 24 different graphs. Each node represents a protein and features such as positional, motif gene, and immunological signature are used with a size of 50 dimensions. Protein-to-protein interactions are represented as edges. Each

Table 11: Model-level explanation on random graph datasets: (a) with 2 or 3-layer GCN, and (b) with 2 or 3-layer GIN. Each task is to classify whether the node belongs to the orbit corresponding task number. The evaluation metric is the *Sub-recall*. We conducted experiments five times and then reported the average and standard deviations. The best performances are shown in **bold**.

(a)

| Task number | 8 | 11 | 16 | 21 | 27 | 31 | 32 | 33 | 35 | 39 | 45 | 47 | 49 | 57 | 59 | 60 | 61 | 62 | 64 |
|---|---|---|---|---|---|---|---|---|---|---|---|---|---|---|---|---|---|---|---|
| Ground-truth orbit | $o_8$ | $o_{11}$ | $o_{16}$ | $o_{21}$ | $o_{27}$ | $o_{31}$ | $o_{32}$ | $o_{33}$ | $o_{35}$ | $o_{39}$ | $o_{45}$ | $o_{47}$ | $o_{49}$ | $o_{57}$ | $o_{59}$ | $o_{60}$ | $o_{61}$ | $o_{62}$ | $o_{64}$ |
| D4Explainer | $0.2 \pm 0.4$ | $0.8 \pm 0.4$ | $0.4 \pm 0.5$ | $0.4 \pm 0.5$ | $0.2 \pm 0.4$ | $0.4 \pm 0.5$ | $0.4 \pm 0.5$ | $\mathbf{0.4 \pm 0.5}$ | $1.0 \pm 0.0$ | $\mathbf{0.6 \pm 0.5}$ | $0.8 \pm 0.4$ | $0.2 \pm 0.4$ | $0.4 \pm 0.5$ | $0.8 \pm 0.4$ | $0.2 \pm 0.4$ | $0.0 \pm 0.0$ | $0.6 \pm 0.5$ | $\mathbf{0.6 \pm 0.5}$ | $0.2 \pm 0.4$ |
| GLGExplainer | $0.8 \pm 0.4$ | $0.6 \pm 0.5$ | $0.8 \pm 0.4$ | $0.6 \pm 0.5$ | $0.8 \pm 0.4$ | $1.0 \pm 0.0$ | $1.0 \pm 0.0$ | $0.8 \pm 0.4$ | $0.8 \pm 0.4$ | $0.8 \pm 0.4$ | $0.8 \pm 0.4$ | $0.6 \pm 0.5$ | $0.6 \pm 0.5$ | $1.0 \pm 0.0$ | $0.6 \pm 0.5$ | $0.6 \pm 0.5$ | $0.8 \pm 0.4$ | $0.0 \pm 0.0$ | $1.0 \pm 0.0$ |
| **UO-Explainer** | $1.0 \pm 0.0$ | $1.0 \pm 0.0$ | $1.0 \pm 0.0$ | $1.0 \pm 0.0$ | $1.0 \pm 0.0$ | $1.0 \pm 0.0$ | $1.0 \pm 0.0$ | $0 \pm 0.0$ | $1.0 \pm 0.0$ | $1.0 \pm 0.0$ | $1.0 \pm 0.0$ | $1.0 \pm 0.0$ | $1.0 \pm 0.0$ | $1.0 \pm 0.0$ | $1.0 \pm 0.0$ | $0 \pm 0.0$ | $1.0 \pm 0.0$ | $0 \pm 0.0$ | $1.0 \pm 0.0$ |

(b)

| Task number | 8 | 11 | 16 | 21 | 27 | 31 | 32 | 33 | 35 | 39 | 45 | 47 | 49 | 57 | 59 | 60 | 61 | 62 | 64 |
|---|---|---|---|---|---|---|---|---|---|---|---|---|---|---|---|---|---|---|---|
| Ground-truth orbit | $o_8$ | $o_{11}$ | $o_{16}$ | $o_{21}$ | $o_{27}$ | $o_{31}$ | $o_{32}$ | $o_{33}$ | $o_{35}$ | $o_{39}$ | $o_{45}$ | $o_{47}$ | $o_{49}$ | $o_{57}$ | $o_{59}$ | $o_{60}$ | $o_{61}$ | $o_{45}$ | $o_{64}$ |
| D4Explainer | $0.8 \pm 0.4$ | $0.6 \pm 0.5$ | $0.6 \pm 0.5$ | $1.0 \pm 0.0$ | $0.6 \pm 0.5$ | $0.8 \pm 0.4$ | $0.8 \pm 0.4$ | $1.0 \pm 0.0$ | $1.0 \pm 0.0$ | $1.0 \pm 0.0$ | $0.6 \pm 0.5$ | $0.4 \pm 0.5$ | $0.8 \pm 0.4$ | $1.0 \pm 0.0$ | $0.8 \pm 0.4$ | $0.8 \pm 0.4$ | $0.2 \pm 0.4$ | $0.4 \pm 0.5$ | $0.6 \pm 0.5$ |
| GLGExplainer | $0.8 \pm 0.4$ | $0.8 \pm 0.4$ | $1.0 \pm 0.0$ | $0.4 \pm 0.5$ | $1.0 \pm 0.0$ | $1.0 \pm 0.0$ | $1.0 \pm 0.0$ | $1.0 \pm 0.0$ | $0.8 \pm 0.4$ | $1.0 \pm 0.0$ | $1.0 \pm 0.0$ | $0.6 \pm 0.5$ | $0.8 \pm 0.4$ | $1.0 \pm 0.0$ | $0.6 \pm 0.5$ | $0.8 \pm 0.4$ | $1.0 \pm 0.0$ | $0.0 \pm 0.0$ | $1.0 \pm 0.0$ |
| **UO-Explainer** | $1.0 \pm 0.0$ | $1.0 \pm 0.0$ | $1.0 \pm 0.0$ | $1.0 \pm 0.0$ | $1.0 \pm 0.0$ | $1.0 \pm 0.0$ | $1.0 \pm 0.0$ | $1.0 \pm 0.0$ | $1.0 \pm 0.0$ | $1.0 \pm 0.0$ | $1.0 \pm 0.0$ | $1.0 \pm 0.0$ | $1.0 \pm 0.0$ | $1.0 \pm 0.0$ | $1.0 \pm 0.0$ | $1.0 \pm 0.0$ | $1.0 \pm 0.0$ | $1.0 \pm 0.0$ | $1.0 \pm 0.0$ |

Table 12: Model-level explanation results on synthetic datasets. The evaluation metric is the Sub-recall. We conducted 5 experiments for each experiment and reported consistent results.

| | BA-Shapes | | | BA-Community | | | | | | Tree-Grid | | | Tree-Cycle |
|---|---|---|---|---|---|---|---|---|---|---|---|---|---|
| | class1 | class2 | class3 | class1 | class2 | class3 | class5 | class6 | class7 | class1 | class2 | class3 | class1 |
| Ground-truth Orbit | $o_{58}$ | $o_{75}$ | $o_{56}$ | $o_{58}$ | $o_{57}$ | $o_{56}$ | $o_{58}$ | $o_{57}$ | $o_{56}$ | $o_{73}$ | $o_{74}$ | $o_{75}$ | $o_{76}$ |
| D4Explainer | $0.4 \pm 0.5$ | $0.6 \pm 0.5$ | $0.4 \pm 0.5$ | $0.8 \pm 0.4$ | $0.2 \pm 0.4$ | $0.8 \pm 0.4$ | $0 \pm 0.0$ | $0.6 \pm 0.5$ | $0.4 \pm 0.5$ | $0.6 \pm 0.4$ | $0 \pm 0$ | $0.2 \pm 0.4$ | $0.8 \pm 0.4$ |
| GLGExplainer | $1.0 \pm 0.0$ | $1.0 \pm 0.0$ | $1.0 \pm 0.0$ | $1.0 \pm 0.0$ | $0.8 \pm 0.4$ | $0.2 \pm 0.4$ | $0.8 \pm 0.4$ | $1.0 \pm 0.0$ | $0.8 \pm 0.4$ | $0.0 \pm 0.0$ | $0.8 \pm 0.4$ | $0.2 \pm 0.4$ | $1.0 \pm 0.0$ |
| **UO-Explainer** | $1.0 \pm 0.0$ | $1.0 \pm 0.0$ | $1.0 \pm 0.0$ | $1.0 \pm 0.0$ | $1.0 \pm 0.0$ | $1.0 \pm 0.0$ | $0.8 \pm 0.4$ | $1.0 \pm 0.0$ | $1.0 \pm 0.0$ | $0.8 \pm 0.4$ | $1.0 \pm 0.0$ | $1.0 \pm 0.0$ | $1.0 \pm 0.0$ |

node is labeled with 121 dimensions of gene ontology sets, allowing for binary classification of 121 categories, similar to a random graph. We pre-trained a GNN on 20 of these graphs and extracted explanations for randomly selected graphs to conduct experiments.

**LastFM-Asia** (Rozemberczki & Sarkar, 2020): This is a social network dataset of LastFM users in Asia. Each node represents a LastFM user in Asian countries, and edges represent the following relationships between them. Node features are composed of artists that users like, which we converted to one-hot encoding for use in experiments. We perform a node classification task to predict the country of each user across 18 countries.

**Gene**: For qualitative experiments, we created a straightforward dataset using the gene network of natural killer cells in humans, as provided by (Zitnik & Leskovec, 2017). Each node of the dataset represents genes, while the features include positional genes, motif genes, and immunological signature information of each gene, with each feature consisting of 50 dimensions. The Molecular Signatures Database (Subramanian et al., 2005) was used to collect the features for each gene. The labels in the dataset indicate whether a given gene belongs to the cell surface receptor signaling pathway, with 1 indicating inclusion and 0 indicating exclusion in the gene ontology set.

## G  LIMITATION AND FUTURE WORK

In this study, we propose UO-Explainer, a GNN explanation method that utilizes graphlets and orbits as explanation units. Though we have demonstrated that UO-Explainer can provide high-quality explanations compared to other baselines on extensive experiments, the motifs we can employ as explanatory units are limited to 2-5 node graphlets. Consequently, we assume thatthis constraint may occasionally hinder the capture of motifs or patterns when crucial patterns are extremely large when the input graph is complex. Since many existing works already can approximate the various shaped subgraphs but do not consider the user-centric perspective, we aim to more focus on the case when prior assumption or knowledge is crucial based on pre-defined units toward human-interpretable explanations.

Table 13: Instance-level explanation results on synthetic datasets. The best performances on each dataset are shown in **bold**. We conducted five experiments for each method by recording the mean and standard deviation.

| | BA-Shapes | | | BA-Community | | | Tree-Grid | | | Tree-Cycle | | |
|---|---|---|---|---|---|---|---|---|---|---|---|---|
| | Sub-recall | Edge-recall | Fidelity | Sub-recall | Edge-recall | Fidelity | Sub-recall | Edge-recall | Fidelity | Sub-recall | Edge-recall | Fidelity |
| GNNExplainer | 0.004 ± 0.008 | 0.616 ± 0.010 | 0.580 ± 0.031 | 0.006 ± 0.008 | 0.491 ± 0.010 | 0.653 ± 0.012 | 0.000 ± 0.000 | 0.629 ± 0.002 | 0.872 ± 0.006 | 0.119 ± 0.000 | 0.699 ± 0.005 | 0.724 ± 0.009 |
| PGExplainer | 0.760 ± 0.025 | 0.915 ± 0.003 | 0.574 ± 0.020 | 0.238 ± 0.015 | 0.667 ± 0.011 | 0.652 ± 0.017 | 0.000 ± 0.000 | 0.647 ± 0.043 | 0.876 ± 0.002 | 0.926 ± 0.051 | 0.992 ± 0.005 | 0.732 ± 0.005 |
| TAGE | 0.682 ± 0.045 | 0.900 ± 0.008 | 0.601 ± 0.016 | 0.352 ± 0.012 | 0.754 ± 0.008 | 0.672 ± 0.010 | 0.003 ± 0.000 | 0.693 ± 0.013 | 0.874 ± 0.004 | 0.963 ± 0.011 | 0.994 ± 0.002 | 0.734 ± 0.004 |
| MixupExplainer | 0.696 ± 0.069 | 0.906 ± 0.012 | 0.612 ± 0.006 | 0.496 ± 0.029 | 0.857 ± 0.019 | 0.693 ± 0.004 | 0.047 ± 0.001 | 0.712 ± 0.001 | 0.877 ± 0.010 | 0.930 ± 0.012 | 0.994 ± 0.013 | 0.734 ± 0.002 |
| SAME | 0.343 ± 0.003 | 0.720 ± 0.009 | 0.547 ± 0.003 | 0.132 ± 0.002 | 0.680 ± 0.010 | 0.642 ± 0.006 | 0.000 ± 0.000 | 0.238 ± 0.013 | 0.846 ± 0.010 | 0.101 ± 0.017 | 0.635 ± 0.023 | 0.692 ± 0.019 |
| EiG-Search | 0.878 ± 0.000 | 0.520 ± 0.000 | 0.605 ± 0.000 | 0.078 ± 0.000 | 0.681 ± 0.000 | 0.695 ± 0.000 | 0.004 ± 0.000 | 0.723 ± 0.000 | 0.885 ± 0.000 | 0.083 ± 0.000 | 0.812 ± 0.000 | 0.703 ± 0.000 |
| MotifExplainer | 0.873 ± 0.069 | 0.890 ± 0.069 | 0.548 ± 0.072 | 0.423 ± 0.045 | 0.714 ± 0.042 | 0.683 ± 0.017 | 0.793 ± 0.046 | 0.857 ± 0.051 | 0.879 ± 0.010 | 0.991 ± 0.000 | 0.993 ± 0.000 | 0.736 ± 0.001 |
| **UO-Explainer** | **0.948 ± 0.016** | **0.984 ± 0.005** | **0.623 ± 0.000** | **0.921 ± 0.083** | **0.970 ± 0.030** | **0.716 ± 0.004** | **0.859 ± 0.032** | **0.900 ± 0.023** | **0.888 ± 0.024** | **1.000 ± 0.000** | **1.000 ± 0.000** | **0.737 ± 0.000** |

Table 14: Instance-level explanation results on real datasets. The best fidelity on each dataset is shown in **bold**. ∗ notation indicates the lower sparsity setting. We conducted five experiments for each method by recording the mean and standard deviation.

| | PPI | | | | | | | | | | | | LastFM Asia | |
|---|---|---|---|---|---|---|---|---|---|---|---|---|---|---|
| | Task0 | | Task1 | | Task2 | | Task3 | | Task4 | | Task5 | | | |
| | Fidelity | Sparsity | Fidelity | Sparsity | Fidelity | Sparsity | Fidelity | Sparsity | Fidelity | Sparsity | Fidelity | Sparsity | Fidelity | Sparsity |
| GNNExplainer | 0.100 ± 0.016 | 0.973 | 0.325 ± 0.004 | 0.999 | 0.417 ± 0.067 | 0.999 | 0.480 ± 0.012 | 0.999 | 0.250 ± 0.001 | 0.999 | 0.331 ± 0.001 | 0.999 | 0.114 ± 0.005 | 0.974 |
| PGExplainer* | 0.023 ± 0.00 | 0.973 | 0.155 ± 0.008 | 0.999 | 0.223 ± 0.077 | 0.999 | 0.101 ± 0.067 | 0.999 | 0.200 ± 0.001 | 0.999 | 0.005 ± 0.000 | 0.999 | 0.011 ± 0.002 | 0.974 |
| TAGE* | 0.031 ± 0.002 | 0.973 | 0.109 ± 0.003 | 0.999 | 0.233 ± 0.052 | 0.999 | 0.138 ± 0.034 | 0.999 | 0.214 ± 0.019 | 0.999 | 0.101 ± 0.020 | 0.999 | 0.086 ± 0.008 | 0.974 |
| MixupExplainer | 0.005 ± 0.000 | 0.973 | 0.002 ± 0.001 | 0.999 | 0.257 ± 0.002 | 0.999 | 0.246 ± 0.003 | 0.999 | 0.246 ± 0.003 | 0.999 | 0.129 ± 0.001 | 0.999 | 0.100 ± 0.001 | 0.974 |
| EiG-Search | 0.180 ± 0.072 | 0.973 | 0.269 ± 0.032 | 0.999 | 0.180 ± 0.053 | 0.999 | 0.379 ± 0.073 | 0.999 | 0.100 ± 0.079 | 0.999 | 0.180 ± 0.072 | 0.999 | 0.095 ± 0.021 | 0.974 |
| SAME | 0.022 ± 0.002 | 0.973 | 0.189 ± 0.003 | 0.999 | 0.194 ± 0.001 | 0.999 | 0.189 ± 0.004 | 0.999 | 0.034 ± 0.005 | 0.999 | 0.129 ± 0.033 | 0.999 | 0.049 ± 0.012 | 0.974 |
| MotifExplainer | 0.070 ± 0.008 | 0.992 | 0.074 ± 0.003 | 0.999 | 0.012 ± 0.028 | 0.999 | 0.129 ± 0.034 | 0.999 | 0.097 ± 0.005 | 0.999 | 0.050 ± 0.021 | 0.999 | 0.085 ± 0.004 | 0.994 |
| **UO-Explainer** | **0.423 ± 0.007** | 0.999 | **0.358 ± 0.022** | 0.999 | **0.425 ± 0.000** | 0.999 | **0.510 ± 0.057** | 0.999 | **0.623 ± 0.029** | 0.999 | **0.413 ± 0.027** | 0.999 | **0.115 ± 0.009** | 0.993 |

