# OpenReview forum: "Toward Human-Interpretable Explanations in a Unified Framework for GNNs"
_ICLR.cc/2025/Conference — Submitted to ICLR 2025_

### Official Review · Reviewer_CDc8 · 2024-11-01

**Soundness:** 2
**Presentation:** 2
**Contribution:** 2
**Rating:** 3
**Confidence:** 4

**Summary:**

The authors propose an graphlet/orbits based GNN Explainer for providing human-interpretable explanations. UO-Explainer decomposes GNN weights into orbits and uses these orbits as explanatory units. It can provide both model-level explanations and instance-level explanations. Experiments on synthetic and real-world datasets show that UO-Explainer outperforms baseline explainability methods in providing meaningful, interpretable explanations.

**Strengths:**

S1: Leverages orbits within small graphlets as human-interpretable units for explanations. This even allows users to define their own units of interest instead of orbits if desired.
S2: Provides explanations in a unified framework at both the model and instance level

**Weaknesses:**

C1: Lack of intuitive representation: what's orbits in graphlet? why orbits are used as explanation unit? What're unique property of graphlet and oribts? What's advantages of graphlet and orbit compared to existing methods? Why the proposed oribit-based explainer can unify both levels?

C2: Limited to using orbits from 2-5 node graphlets, restricting the explanation patterns. There may be cases where important patterns in the data involve larger graphlets that cannot be captured by this restricted set of units.


C3: As a follow-up, we need to predefine the graphlets/orbits, rather than learning them from data. Would it be possible to extend the base set of graphlet. For example, considering these 2-5 node graphlet as seeds, how to dynamically discover larger graphlet that might be more meaningful to exaplain the prediction ability of a GNN.

**Questions:**

Please refer to weakness

---

### Official Review · Reviewer_J3Jj · 2024-11-03

**Soundness:** 2
**Presentation:** 2
**Contribution:** 3
**Rating:** 5
**Confidence:** 5

**Summary:**

This submission introduces a new l framework for explaining Graph Neural Networks (GNNs) that provides both model-level and instance-level explanations in a unified manner. UO-Explainer leverages graphlets and their orbits—small, connected, non-isomorphic subgraphs—as human-interpretable units to decompose GNN predictions. The framework enables users to utilize prior knowledge by selecting relevant graphlets for detailed, interpretable explanations. UO-Explainer decomposes class-specific model weights into orbit units, facilitating the identification of important substructures in both general and instance-specific contexts. Extensive experiments on synthetic and real-world datasets demonstrate that UO-Explainer outperforms existing methods in delivering high-quality explanations.

**Strengths:**

1. The idea is interesting and promising in unifying model-level and instance-level explanations for GNNs.
2. The technical approach is novel, incorporating orbit basis learning and class-orbit score learning.
3. The paper is well-organized, with self-contained figures that aid understanding.
4. The experiments are comprehensive, covering 8 node classification datasets.

**Weaknesses:**

1. The title and abstract overstate the scope, as the method can only explain node classification tasks.

2. The method’s reliance on all subgraphs with up to 5 nodes as explanation units does not inherently ensure meaningful or domain-relevant structures. It is unclear how this improves human interpretability compared to subgraph-based methods that use constraints like connectivity.

3. In real-world datasets such as Gene, linking small graphlets to functionally significant groups can be problematic. The approach does not consider node features or types, making it difficult to distinguish substructures with different real-world meanings or implications. For instance, C-C and CO substructures correspond to the same graphlet, which poses an issue.

4. The notations need improvement, as the notation for the downstream layer does not consider activation functions.

5. The reference paper mentions two fidelity measurements: Fidelity+ and Fidelity-. This submission uses Fidelity+, and I suggest that the authors specify this clearly.

6. Add robust fidelity measurements. As recent studies [1, 2, 3] have shown, Fidelity in the graph domain suffers from out-of-distribution (OOD) issues. Specifically, in Eq. 10, $f_{\text{prob}}()$ is trained on datasets containing entire graphs, whereas  $G_{vi} - G_{vi}^{\text{ex}}$ is a smaller subgraph with a different distribution. Thus, the prediction  $f_{\text{prob}}(G_{vi} - G_{vi}^{\text{ex}})$ may not be reliable. I recommend that the authors consider robust fidelity measurements, such as Robust Fidelity [1], OAR, SimOAR[2], or GInX-Eval[3], or demonstrate that this distribution shift does not impact the main results.

[1] Zheng, Xu, et al. "Towards Robust Fidelity for Evaluating Explainability of Graph Neural Networks." The Twelfth International Conference on Learning Representations. (2024)
[2] Fang, Junfeng, et al. "Evaluating post-hoc explanations for graph neural networks via robustness analysis." Advances in Neural Information Processing Systems 36 (2023).
[3] Amara, Kenza, Mennatallah El-Assady, and Rex Ying. "GInX-Eval: Towards In-Distribution Evaluation of Graph Neural Network Explanations." XAI in Action: Past, Present, and Future Applications. 2023

**Questions:**

See above.

---

### Official Review · Reviewer_cpes · 2024-11-05

**Soundness:** 3
**Presentation:** 2
**Contribution:** 2
**Rating:** 5
**Confidence:** 4

**Summary:**

The paper introduces UO-Explainer, a framework for Graph Neural Networks (GNNs) that provides human-interpretable explanations at both model and instance levels. It utilizes graphlets and orbits as interpretable units to reveal the significance of specific graph structures for predictions. This unified approach aims to address limitations in previous GNN explainability methods, particularly in providing coherent, interpretable insights. The experiments on various synthetic and real-world datasets demonstrate its effectiveness.

**Strengths:**

1. The idea of leveraging graphlets and orbits as units for interpretation is interesting

2. The experimental results show UO-Explainer accurately provides explanations than baselines at both model-level and instance level.

3. The source code is publicly available

**Weaknesses:**

1. The paper aims to provide human interpretable explanations. However, there are no experiments asking human evaluators to evaluate the quality of the explanations. The authors should consider adding some experiments of human evaluation.

2. The model-level explanation is only evaluated on synthetic datasets. It is unclear if such kind of model-level explanation really makes sense on real-world datasets. For example, does there really exist a model-level graphlet for each class for molecular graphs, which is able to explain the characteristics of each class captured by the target GNN model?

3. Though the idea of leveraging graphlets and orbits as units for interpretation is interesting, the authors might need to give more explanations and real-world examples showing why they make sense in real-world.

4. In lines 201-202, this paper introduces a model-level explanation by decomposing class weights into a linear combination of orbit bases. However, I think it will potentially lose some contextual information between the graphlets, because these bases only capture local information, and they might miss out on broader contextual patterns in the graph. For example, some predictions might depend on interactions between distant nodes or on the overall graph structure, which cannot be encapsulated by small, isolated graphlets.

4. In line 259-261, the paper introduces the instance-level explanation by decomposing the prediction value of the target node into orbit units. But the motivation of the proposed method is not clearly introduced.

5. I am concerned about the efficiency. To generate the instance-level explanations, the proposed method needs to go through all graphlets that include all 0-72 orbits with time complexity $O(|O||V| d^{k−1})$ based on Algorithm 3 and 4. This search method is too expensive to extend to large-scale graphs. Moreover, this paper lacks an experimental analysis of the time complexity. A running time comparison is highly suggested.

**Questions:**

Please see above weaknesses

---

### Official Review · Reviewer_oD7a · 2024-11-05

**Soundness:** 2
**Presentation:** 3
**Contribution:** 2
**Rating:** 3
**Confidence:** 5

**Summary:**

The paper addresses the problem of explainability in (black-box) graph neural networks (GNNs). As the existing methods lack explanations that are human interpretable as well as a unified framework to perform both instance specific and model-level explanations, the paper proposes a framework based on graphlets (and orbit bases). These predefined graphlets and their associated orbits contribute to both instance and model-level explanations.

**Strengths:**

- The problem of explanations of GNN is relevant and timely as GNNs are being applied in many domains.

- The unified framework of having both instance-level and model-level explanations is interesting.

- The experiments have many different settings. The number of baselines and datasets is comprehensive.

**Weaknesses:**

- The graphlets (orbit bases) as human interpretable units need justifications.

- A strong assumption is that both the instance-level and model level explanations depend on these graphlets (orbit bases). This also needs justification.

- Some experimental settings could be improved.

**Questions:**

1. Why are the graphlets (orbit bases) important? Why are they human interpretable? One could actually perform a user study to justify this. Do you have such evidence?

2. Why is the model-level explanation a linear combination of the orbit bases? What if the model-explanation depends on completely different factors?

3. The method seems to be restricted to only node classification. How will it generalize to other tasks (e.g. graph classification)?

4. Maybe I am missing something: how are you evaluating the baselines in the same framework especially for the model-level explanations?

5. Could you emphasize on the sub-recall measure? The model-level explanations are being evaluated only on that. Why is it the case? Why aren't the measures of GLGExplainer being used?

6. Using only synthetic data for the mode-level explanations needs justification.

---

### Meta-Review · Area_Chair_xoWY · 2024-12-21

**Metareview:**

The paper proposes an algorithm for explaining black-box GNNs. The paper is motivated by the observation that existing methods lack explanations that are human interpretable and the ability to perform both instance specific and model-level explanations. The paper proposes a framework based on predefined graphlets and their associated orbits to perform both instance and model-level explanations. The reviewers have highlightes several concerns related to the justification of using graphlets, their connections to human interpretability, and substantiating the various claims made in the paper. The authors opted not to submit a rebuttal, leading to the conclusion that the paper is not yet ready for publication.

**Additional Comments On Reviewer Discussion:**

The authors did not post a rebuttal.

---

### Decision · Program_Chairs · 2025-01-22

Reject